# Gradual Loss of Social Group Support during Competition Activates Anterior TPJ and Insula but Deactivates Default Mode Network

**DOI:** 10.3390/brainsci13111509

**Published:** 2023-10-25

**Authors:** Burcu Ozkul, Cemre Candemir, Kaya Oguz, Seda Eroglu-Koc, Gozde Kizilates-Evin, Onur Ugurlu, Yigit Erdogan, Defne Dakota Mull, Mehmet Cagdas Eker, Omer Kitis, Ali Saffet Gonul

**Affiliations:** 1School of Nursing and Midwifery, La Trobe University, Melbourne, VIC 3086, Australia; b.ozkul@latrobe.edu.au; 2SoCAT Lab Department of Psychiatry, School of Medicine, Ege University, Izmir 35080, Turkey; cemre.candemir@ege.edu.tr (C.C.); eroglusedaa@gmail.com (S.E.-K.); yigit_gnr@hotmail.com (Y.E.); mehmet.cagdas.eker@ege.edu.tr (M.C.E.); 3International Computer Institute, Ege University, Izmir 35100, Turkey; 4Department of Computer Engineering, Izmir University of Economics, Izmir 35330, Turkey; kaya.oguz@ieu.edu.tr; 5Department of Psychology, Faculty of Letters, Dokuz Eylul University, Izmir 35390, Turkey; 6Neuroimaging Unit, Hulusi Behcet Life Sciences Research Laboratory, Istanbul University, Istanbul 34093, Turkey; gozde.kizilates@gmail.com; 7Department of Neuroscience, Aziz Sancar Institute of Experimental Medicine, Istanbul University, Istanbul 34093, Turkey; 8Department of Fundamental Sciences, Faculty of Engineering and Architecture, Izmir Bakircay University, Izmir 35665, Turkey; onur.ugurlu@bakircay.edu.tr; 9Department of Neuroscience, Health Sciences Institute, Ege University, Izmir 35080, Turkey; 10Department of Radiology, School of Medicine, Ege University, Izmir 35080, Turkey; omerkitis@yahoo.com

**Keywords:** default mode network, social group, social support, salience network

## Abstract

Group forming behaviors are common in many species to overcome environmental challenges. In humans, bonding, trust, group norms, and a shared past increase consolidation of social groups. Being a part of a social group increases resilience to mental stress; conversely, its loss increases vulnerability to depression. However, our knowledge on how social group support affects brain functions is limited. This study observed that default mode network (DMN) activity reduced with the loss of social group support from real-life friends in a challenging social competition. The loss of support induced anterior temporoparietal activity followed by anterior insula and the dorsal attentional network activity. Being a part of a social group and having support provides an environment for high cognitive functioning of the DMN, while the loss of group support acts as a threat signal and activates the anterior temporoparietal junction (TPJ) and insula regions of salience and attentional networks for individual survival.

## 1. Introduction

Living beings (including bacteria) form groups to overcome environmental challenges that cannot be managed individually [1]. Group-forming behaviors become more complicated with the development of species’ cognitive abilities as they climb the phylogenetic tree’s upper branches. So far, humans have created functional groups of different sizes and co-operation levels for their good and survival [2,3]. Being part of a group starts in childhood and is effective throughout life so that even people wearing the same color t-shirts feel closer to each other [4].

People come together on many occasions, but these gatherings are generally transient; bonding and support within the groups are weak. Social groups consist of individuals who recognize themselves as members of the same social category who share similar interests, values, and representations. When the individuals categorize themselves as part of such a group, their self-concept shifts from the individual (“I” or “me”) to the group level (“we” or “us”) [5]. The group’s priorities, which are echoed in members’ affect and behavior, become more important than the individual’s priorities [6,7]. Individuals who behave fairly in general, act in a biased way toward in-group when against out-group, especially in zero-sum situations [8]. At the same time, members expect reciprocity for their supportive behaviors [9]. In the case of not obtaining enough support, members may feel ostracism, frustration, loss of self-esteem and confidence, similar to the symptoms of depression, especially in challenging situations [10,11,12]. One fundamental result of ostracism is the shifting of the individual’s self-concept from the collective/group (“we” or “us”) to the individual (“I” or “me”) level, which necessitates taking steps for individual survival [13]. Therefore, understanding how the brain responds to the social support loss that changes self-perception and subsequently tries to adapt to the less predictable and unsafe environment helps us comprehend the neurophysiological foundations of resilience.

In recent years, with the help of meta-analyses and meta-analytic tools such as Neurosynth, researchers have started to better understand the neurobiological bases of higher cognitive functions such as the perception of one’s self and their relation to environmental changes (https://neurosynth.org/, accessed on 8 October 2023). Those higher cognitive functions include many domains of cognitive processes such as self-processing, social processing or theory of mind as well as conflict monitoring [14]. The cognitive functions are the basis of the ‘self’ function, which link personal identity and experience in the functionally correlated structures known as the default mode network (DMN) [15]. The core DMN consists of the medial prefrontal cortex (mPFC) and posterior cingulate cortex (PCC)/precuneus [16,17]. The inferior parietal cortex (IPC)/temporoparietal junction (TPJ), medial temporal region, and anterior temporal cortex are the other nodes that accompany core DMN nodes depending on the operations being executed [18]. The mPFC plays an essential role in self-referential processing, while the PCC/precuneus contributes to self-related episodic memory retrieval and integrating the information from cortical association areas [19,20]. Self-referential processes accept oneself as the central dimension through self-monitoring and the ability to experience a congruent observed emotion [21,22]. Recent studies showed that the mPFC of a DMN is also active during neural processes representing “group-self” which suggests that the neural networks representing “I” and “we” overlap considerably in the current low-resolution fMRI images [5,23,24,25]. One reason for the overlap is that the neural process of self and familiarity take place in the same or close regions at the mPFC [21,26,27]. The latter organization might help mPFC with IPC/TPJ to infer and discriminate one’s own and the other’s mind (theory of mind, ToM), which is crucial for group formation [28]. Therefore, studies suggest that both individual- and group-self functions take place in nodes of the DMN. 

It has been shown that subjects who were excluded during multi-person games had increased activity in the dorsal anterior cingulate (dACC) and anterior insula (AI), correlated with social distress [29,30] but see [31]. Social distress is an alarm for the discrepancy between what is expected and what is obtained and a warning for possible loss of social support or ostracism. The AI and dACC are parts of the general neural system (salience network (SN)) responsible for threat detection and processing [29]. However, these findings came from the studies that (1) used minimal group paradigms, which have low ecological validity for representing social groups by discounting the effect of bonding, and the necessity of support from the group for achievements [32], (2) neglect self-perception of members before the ostracism (3) have a rapid exclusion process which did not allow investigation of the switching from one social status to another (e.g., in-group to out-group).

The current study created an experimental environment where subjects need real-life friends’ support for wins in a social competition. Initially, subjects had a predictable and secure environment with the support (biased responses) as expected. However, as the game proceeded, the subjects had fair responses (loss of supportive environment) and then biased responses for their rivals (insecure environment) in the competition. This experimental design helped us understand how the brain responds to gradual social support loss that changes self-perception and subsequently tries to adapt to the less predictable and unsafe social environment. We hypothesized that having or losing group support would significantly affect the brain’s functional circuits in a challenging situation. We preferred real-life social groups because the bonding among members is stronger than minimal groups [33,34]. Therefore, we expected more robust neurological responses compared to previous studies.

Although the task could activate multiple brain function domains (e.g., visual computation or ranking), we focused on the brain activity associated with group (social) support and its loss. Therefore, we specifically followed signal changes during the feedback stages of the task, when the subjects saw the jury scores, which their friends determined. 

## 2. Materials and Methods

### 2.1. Experimental Model and Subject Details

#### Participants

After receiving approval from the Institutional Ethics Committee for Medical Studies (18 August 2014, 14-7/17), we recruited the participants via fliers or emails from among the students of our university. The fliers or electronic invitations were looking for volunteer groups consisting of four friends for a competition between home and a rival university in a guessing game. More than 50 groups had applied to the study; however, only 20 groups (80 university students) met the study criteria. Two groups were dropped due to compliance problems during the fMRI scan or jury duty. The final 18 groups, including 72 participants in total, comprised the study sample.

All participants within the groups were evaluated separately. At the initial interview each applicant described themselves as good or close friends with others within the group and they had known each other for over one year and gave at least seven points out of 10 to their friendship. Each participant completed a Rosenberg self-esteem scale [35]. At the end of initial assessment, a subject from the group was assigned for the competition (total 18 subjects; 10 females, mean age: 21.7 ± 1.6 years) and the other three members joined the jury. Although it was said that the selections were random, the competitor subjects were selected according to inclusion criteria, being at the age between 18–25 years old and right-handed. Exclusion criteria included (1) a history of present or past psychiatric illness, (2) unstable medical disease (e.g., diabetes mellitus, hypertension, etc.), (3) any first degree relative with bipolar or psychotic disorders, (4) a history of head trauma with loss of consciousness, and (5) low self-esteem score obtained from the Rosenberg self-esteem scale (<15) (to reduce the risk of rapid demoralization in the latter part of the game when the subjects must compete without support).

### 2.2. Method Details

#### 2.2.1. Psychometric Assessment

All participants were briefly interviewed for sociodemographic background and then screened with SCL-90 for all possible psychiatric symptoms. The chosen competitors who met the inclusion/exclusion criteria completed the multi-dimensional scale of perceived social support, and the inventory of socially supportive behaviors (ISSB) [36] scales before the fMRI game session. All group members had also filled in a need–threat scale [37] after the fMRI scan but before the group members came together.

#### 2.2.2. The Game

The same instructor (BÖ) informed all group members that they would compete against a rival group from another local university in a guessing game to observe the effect of university education on (improving) students’ visuospatial abilities. The instructor described the game as follows: one of the group members would be chosen randomly to compete in fMRI while the others would become jury members in separate rooms. The competitor subject is asked to guess whether the number of squares on the screen in the fMRI is higher or lower than the presented number (range: 30–50) on a screen. The images were random and different for the subject and the rival (Figure 1). The instructor warned that if their response time exceeds 3 s, or they do not decide on purpose, the computer would randomly pick an answer. They were also informed that the team would be disqualified automatically if they missed more than three consecutive responses. After the competitors’ response, each juror would see the screens of both competitors (subject and rival). The jury had 10 points in each trial and were expected to split these points between their competitor friend (subject) and their rival, such as 8/2, 7/3, 6/4 or 5/5, according to the difficulty level of each image and the answers of the competitors. Thus, the jury has the freedom of giving high points to a competitor for an incorrect answer with a very difficult image. At the end of the trial, the competitors see both their own and rivals’ images with a green tick (correct guess) or a red cross (wrong guess) and the juror’s decisions for six seconds. This way, the competitor can compare the difficulty levels of their own and rival’s images and performance, and evaluate the decisions of jury members. However, the jury members could not see the final scores and thus independently scored each trial without knowing the decisions of other jury members or the scores of the competitors. The instructor further explained that the competitor subjects would receive an amount of money based on the jurors’ points. Groups were also informed that the members would share a sum of money to compensate for their time (approx. 25 USD). Depending on their total score, they might increase their earning up by 50%.

After all members had reported that they understood the task, one subject from each group went to the fMRI room to compete with the rival via playing the game while the rest went to separate rooms for jury duty. None of the members had contact until the end of the game. Before getting onto the scanner, the competitor subjects were informed that one of the rival group members could not join their friends due to personal reasons, but the rest still wanted to participate. The instructor also specifically explained the advantage of this situation to the subjects because three of the five jurors are friends of the subject.

Although the rival university’s name was real, all the other mentioned characters were fictitious. All the juror computers and the one that used for presentation in fMRI were synchronized and the game started at the same time. During the game in fMRI, the jury points and the performance of the subjects were predetermined to create a gradually declining social support environment, independent from the actual performance of the subjects and jury points.

#### 2.2.3. Stimuli and Apparatus

At the beginning of each trial, the subjects in fMRI were asked to guess the number of squares in the presented image made up of various geometric shapes, such as varying numbers of squares, hexagons, circles, and stars. Each image was designed to have 30, 40, or 50 squares with varying numbers of other shapes. The participants decided whether the number of squares was higher/lower than the given number by pressing the left/right button. The reason behind such a complex task is so that the competitor subjects cannot be sure about the accuracy of their answers and we can manipulate his/her performance. During the task design a pilot study with 30 university students was performed to determine the difficulty levels of each created image. One hundred and thirty pairs of images with different numbers and distributions of shapes that were obtained by using different combinations of 10 generated image groups were rated. We presented two adjacent images to the students and asked to assign difficulty scores, such as 8/2, 7/3, 6/4, and 5/5 based on the difficulty of predicting the number of squares. For example, 8/2 means it is very difficult to guess the right number of squares in the first image and very easy to guess right in the second one. Then, the score difference between the two images and the mean scoring of each image was calculated. Image pairs were degreed as high difficulty (score difference < 1, similar to predicting the number of squares) to low difficulty (score difference > 3, easy to guess the number of squares of one image compared to the other image). After the assessment process, we decided to use 75 image pairs. The images with high and low difficulty levels were then distributed to four guessing feedback situations: (1) the competitor and rival were both correct; (2) the competitor—correct, the rival—incorrect; (3) the competitor—incorrect, the rival—correct; (4) both the competitor and the rival were incorrect. The ratio of these situations was also determined before the experiment (Appendix A).

All stimuli were presented by a desktop computer screen projected to MRI-compatible monitor reflecting on a mirror over the head coil in the scanner. The participants responded via a response grip placed under both their thumbs. The presentation 19.0 (http://www.nitrc.org/projects/presentation, RRID: SCR_002521, accessed on 18 December 2014) was used to show the stimulus.

#### 2.2.4. The fMRI Task Protocol

The fMRI task was composed of 75 trails. Each trial has the following six stages (Figure 1):

*Baseline:* Each trial starts with the baseline screen with a fixation crosshair for three seconds (stage 1)

*Computation:* In the first screen, the image with different numbers of geometrical shapes is presented where the competitors are expected to guess the number of squares in the picture in three seconds (stage 2). Then, in the second screen, the subject should have chosen the appropriate answer on whether the number of squares is more or less than the presented value on the screen in three seconds (stage 3). 

Anticipation: The subjects wait for six seconds with an hourglass screen for the jury’s response (stage 4).

Feedback: After the waiting period, the subjects see both their own and rivals’ images with a green tick (correct guess) or a red cross (wrong guess), and the jury’s decisions for six seconds. This way, the competitor can compare the difficulty levels of their own and rival’s images and performance, and evaluate the decisions (biased or fair) of the jury members. The points are the mean score of all jurors’ decisions with a majority of the competitor’s friends assigning the final score in each trial (stage 5). 

Status: As the final screen of the trial, the money bars, which represent the total amount the subject and the rival are earned, are presented for three seconds (stage 6).

The total experiment was divided into three phases, each including 25 trials. The subject had significantly more points in 80% of trials in the first 25 trials of the game (high-support phase, HSP); in the next 25 trials, the subject had more points in 48% of the trials (fair phase, FP). During the last 25 trials, subjects received more points in only 20% of the trials (ostracism phase, OP). As the participants felt the support of their friends at a higher level in the images with high difficulty, these images were used at a higher rate. All images were numbered and recorded with their presentation order in the experiment. The total duration of the task was 30 min and 42 s. After the game, before the group come together, the subjects had a debriefing session with a trained psychologist or nurse to reduce their frustration. During the debriefing subjects did not say or imply that they were suspicious of the game, and they explicitly said that they were feeling frustrated about the results. The debriefing session focused on misunderstandings and possible technical errors which might mislead the subject or jury.

### 2.3. Quantification and Statistical Analysis

#### 2.3.1. Behavioral Analysis

We used the Wilcoxon test to compare the differences between the high-support phase and ostracism phase with the belonging, self-esteem, meaning of existence, and control subscales of need-threat scale. The Wilcoxon test was also used to compare the differences between their emotions (motivation, happiness, sadness, fear, anger) before the study and after the study (Appendix A). Data analysis was performed using SPSS 22.0 (https://www.ibm.com/products/spss-statistics, accessed on 10 June 2023, RRID:SCR_019096).

#### 2.3.2. fMRI Image Acquisition

Imaging sessions were performed using a 3T Siemens Magnetom Verio Scanner (Syngo MR B17, Erlangen, Germany) with a 12-channel head matrix coil. Firstly, structural images were acquired with T2-weighted axial TSE and coronal 3D-SPACE FLAIR (Dark Fluid) for any possible pathology and T1-weighted 3D- MP-RAGE sequence (TR/TE 9.9/2.9 ms; matrix size 256 × 256 mm; 1 × 1 × 1 mm resolution, axial orientation) for coregistration with functional images. Then, fMRI data were collected using T2*- weighted Echo-planar imaging (EPI) sequence (TR/TE 1980/32 ms, 22 sequential axial slices, whole brain, 3.5 mm slice thickness, 1 mm gap, resulting voxel size 3.125 × 3.125 × 4.5 mm, matrix 64 × 64 voxel, 200 mm field of view, 70° flip angle).

#### 2.3.3. Preprocessing

Preprocessing functional EPI images was performed using SPM12 (Statistical Parametric Mapping, Wellcome Trust Centre for Neuroimaging, London, UK, https://www.fil.ion.ucl.ac.uk/spm, RRID:SCR_007037, accessed on 4 October 2023) implemented in MATLAB R2019a (https://www.mathworks.com/campaigns/products/, RRID:SCR_001622, accessed on 4 October 2023). First, raw DI-COM files were converted to NIFTI images with a built-in conversion tool in SPM12. All converted images were then reoriented and preprocessed, including realignment, co-registration, slice-timing, segmentation into grey matter (GM), white matter (WM), cerebrospinal fluid (CSF), normalization to a template space of MNI (Montreal Neurological Institute). During the realignment step, all images were controlled for excessive head movements. The deviations bigger than the voxel size were discarded before further analysis. The small movements were corrected with rigid body transformation according to the reference image. Before using functional EPI images in the statistical model, all normalized images were smoothed with an 8-mm Gaussian Kernel.

#### 2.3.4. Univariate Analyses

All first-level and second-level analyses of fMRI data were performed with SPM12. It was assumed that the measurement levels were independent and had unequal variances. The absolute threshold mask was defined as including voxels with a GM or WM value greater than 0.2. 

As a second-level group analysis, a 2 × 3 flexible-factorial model was designed. The model design consists of two factors: (1) task and (2) phase. The task involves two stages (baseline and feedback), whereas phase involves three levels (HSP, FP and OP). All subjects were included in the flexible-factorial design, where six contrast images were assigned per all subjects. The main effects of the baseline and feedback factors, and interactions between the phase levels (i.e., HSP, FP, and OP) were analyzed with the appropriate contrast vectors.

#### 2.3.5. Contrasts

To observe the hemodynamic amplitude in response for each stage in the task, all possible contrasts were constituted for each level of the game for each subject. Forming all contrast, i.e., assigning weights to each beta value (*β*), gives the opportunity to answer the question of which voxels in one stage respond significantly different than to the other stage.

The first-level analysis, i.e., individual analysis for each subject, the contrast vectors were constituted based upon the following model:(1)$$\gamma_{n}=c_{i}*{\hat{\beta }}_{j},i=\left[1,15\right],\;j=\left[1,3\right],\;n=\left[1,18\right]$$

Here, $c_{i}$ represents the contrast coefficients, ${\hat{\beta }}_{j}$ represents the stages of the task and *n* shows the subject number.

Hence, 15 contrast vectors were generated for three levels of the five conditions in total for each subject. Afterwards, the related contrast images of the subjects were used in the second-level factorial design for further analyses.

For the second-level analysis, the contrast vector of the model is given as follows:(2)$$\gamma_{i}=c_{i,j}*{\hat{\beta }}_{i,j},i=\left[1,5\right],\;j=[1,3]$$

Here, *c* is the contrast matrix, which consists of the *β* weights, while *j* represents the three levels of the task and *i* represents the stages of level. For instance, for the first level, task stages and *β* values are modeled as follows:(3)$$\gamma_{1}=c_{Base,1}*{\hat{\beta }}_{Base,1}+c_{Comp,1}*{\hat{\beta }}_{Comp,1}+c_{Anticip,1}*{\hat{\beta }}_{Anticip,1}+c_{Feedb,1}\;*{\hat{\beta }}_{Feedb,1}+c_{Status,1}*{\hat{\beta }}_{Status,1}$$

The above equation can be used to determine the subjects’ responses in different levels of different combinations of the stages. For both first- and second-level analysis, the contrasts which exhibit the most significant results were determined as *Feedback_HSP_ <> Feedback_FP_ <> Feedback_OP_*.

For the representation of the BOLD signal changes in time, in other words, time series analysis of the BOLD signals, and the correlations with the need–threat scale, beta values were also extracted for each subject. This allows us to test the hypothesis from another point of view (see beta values).

We performed whole brain analyses and reported clusters with more than five contiguous voxels with a global-maxima meeting with a family-wise error (FWE)-corrected threshold of *p* < 0.05.

#### 2.3.6. Beta Values

For further analyses, each subject’s mean beta values were also generated one-by-one. To find the beta values of the model for each determined contrast, MarsBaR toolbox 96 (RRID:SCR_009605) that runs on SPM is used (http://marsbar.sourceforge.net/, accessed on 4 October 2023). The first step of extracting these values is generating the ROI mask. The center of the mask is that given at the MNI co-ordinates of clusters obtained from the aTPJ and AI. Once the ROI mask is created, beta values are imported into the MarsBar toolbox to extract the parameter estimates. Then, the mask is converted into the MarsBar ROI object (i.e., a .mat file). The mean beta values are extracted over the entire ROI, from the contrast vectors obtained from the first-level design, and from the design matrix. This procedure was followed for each subject. Once the parameter estimation was performed, it is possible to achieve all beta values for each subject and group averages for the desired stage.

The extracted beta values were used to correlate the behavioral analyses with the belonging, self-esteem, meaning of existence, and control subscales of the need–threat scale, as well as self-esteem and perceived social support in SPSS 22.0 (https://www.ibm.com/products/spss-statistics, RRID:SCR_019096, accessed on 10 June 2023).

## 3. Results

### 3.1. Psychometric Results 

All the sociodemographic and psychometric characteristics of the subjects were presented in Appendix A. During the post-game interviews, all subjects described the initial part of the game as the winning period, and the last part of the game as the losing period. They reported that their support started to decrease in the middle of the game. Compared to the winning period, subjects reported reduced scores in all subscales of the need–threat scales (belonging, Z = 3.1 *p* = 0.002; self-esteem, Z = 3.4 *p* = 0.001; meaningful existence; Z = 2.91 *p* = 0.003; control, Z = 2.8 *p* = 0.004, Appendix A), suggesting that the task had a significant effect on the perception of their self-esteem and group (social) support. The reaction times of the competitors decreased according to phases of the study (HSP (1395 msec), FP (1220 msec), and OP (1167 msec)). (F = .12.2 df = 1,17 *p* = 0.003).

### 3.2. fMRI Results

In the fMRI analyses, we focused on the feedback stage of the game to observe the effects of social support from friends. We presented the BOLD signal changes at the fair phase (FP) and ostracism phase (OP) based on the high social support phase as the social support decreases gradually (Figure 2).

We observed that during FP compared to HSP when the support (bias answers) decreased to fair levels, the BOLD signal decreased in the DMN areas (medial PFC, PCC and angular gyrus—inferior parietal cortex (IPC/pTPJ)) and middle temporal cortex, while there was an increased BOLD signal in the bilateral supramarginal gyrus (anterior TPJ), left dlPFC and visual cortex (Table 1, Figure 2a and Figure 3A). In the OP, as the social support decreased, the BOLD increased in the supramarginal gyrus extending to dorsal parietal regions and visual cortex (Table 2, Figure 2b). Furthermore, we observed activation in the anterior insula and ventral striatum, dlPFC at BA 44 and 10. DMN activity remained low compared to HSP (Figure 2b). The most significant change from FP to OP is increased activity in the anterior insula at BA 13 (Table 3, Figure 3B). Among the need–threat subscales after the game scoring, right AI activity showed a negative association with belonging (r = −0.57 *p* < 0.05) (Figure 3B).

## 4. Discussion

The current study provided empirical data on how BOLD signals change gradually in brain areas that are from different networks during social group support loss. At the beginning of the game, the friends from the same social group in the jury provided easy wins (a safe and predictable environment) by giving biased points (support). It was an expected situation because biased behaviors towards in-group members is a (social) norm in many groups. However, we observed decreased DMN activity as the game proceeded and the subjects received fair support. Furthermore, we detected increased activity in the aTPJ (supramarginal gyrus). In the last part of the game (OP), subjects had low support from the friends in the jury and BOLD activity increased in the insula, VS (caudate), aTPJ and attention-related areas dlPFC and dPC.

DMN is associated with a variety of internally directed mental representations such as self-reflection, autobiographical memory, future event simulation, mind wandering, and conceptual processing, which are advanced forms of thoughts that need a vast amount of information and computational resources [18,34,38,39]. Common features of all these mental processes is the use of available and previously processed information and needing minimal environmental sensory stimulation. In environments that are easy to foresee, mental activity can minimally engage with the actual sensory environment and shift to hierarchically high-order cognitive functions without requiring any further new information from the environment [38,40,41]. Our results suggest that during the social group support, which provided a predictable and positive affective setting for the subjects, DMN activity was present despite ongoing competition.

It was recently suggested that evaluating a low threat level with a predictable environment is a functional necessity of DMN for its multi-domain integrative functioning [38,42]. Consequently, DMN carries out some form of probabilistic estimation of past, future, and hypothetical events, which might be helped by a follow-up in a predictable environment by minimizing prediction error [38,43,44]. The broken prediction error minimization on environmental inputs would reduce the DMN activity while activating other networks such as salience and attention networks [45,46]. The right IPC/pTPJ region (angular gyrus) of DMN is a candidate region for prediction error signaling in detecting environmental changes, including changes in abstract environments such as the social environment. Therefore, it is not surprising to observe that the right IPC/pTPJ plays a crucial role in executing ToM functions such as predicting other peoples’ intentions [28]. It should be noted that, with its ability for prediction error minimization, this region might be a node for more than one functional activity of DMN [44]. Thus, the higher BOLD activity in pTPJ (angular gyrus) and mPFC at the beginning the game might be associated with following and predicting the intentions and behaviors of fellow jury members. The later decreased BOLD activity in these regions might be associated with a broken prediction error signal. However, it should be noted that reduced DMN activity in FP does not show reduced functional connectivity within the DMN. Indeed, other studies focused on the DMN connectivity during ostracization (including the one whose data we used in the current study) showed increased connectivity within the network [31,47,48].

As the game proceeds, independent from the subjects’ performance, the biased points (support) decreased at fair levels. Hence, the initial safe and predictable environment by group norms changed to an unpredictable environment. The BOLD activity reduced in the core DMN while it increased in the aTPJ and later AI. The aTPJ is located anteroventrally to the IPC/pTPJ of DMN. The functional and structural connectivity of this region is still under investigation. Some studies suggested that it is a part of the salience network [49,50], while others suggest it is a part of the frontoparietal attention networks (FPN) [17]. In contrast to pTPJ of DMN, which is sensitive to predictable stimuli, the aTPJ is sensitive to unpredictable, infrequent but potential behavioral relevance stimuli that need evaluation by high-order systems [51,52,53]. The stimuli should also be salient and recognizable for bottom-up attentional reorientation. Therefore, the aTPJ might be a crosstalk region of both attentional and salience networks [49]. Our findings support this view; the aTPJ became activated with insula (SN) by unpredictable feedback (stimuli) and reorients incoming stimuli for further high-order top-down processing. This processing might facilitate the FPN (dorsal parietal regions) and visual system activation for the incoming stimuli.

Our finding of increased AI was in line with previous studies in which an association between the AI and aversive emotional experiences during exclusion or expectation violation was found (significant difference between what is expected and what is obtained) [54]. AI activation is not specific for aversive feelings such as disgust, but also observed in the subjective feeling of all bodily sensations and the higher-order integrations of those sensations such as self-awareness and time-perception [55]. The AI is coupled with the pre- and subgenual cingulate cortex, is a part of the salience network and has robust functional connectivity with limbic structures important for emotion, reward, and homeostatic regulation [50]. Our finding of increased AI activity as a part of SN responded to the loss of biased points (supportive behavior), which was a salient stimulus to the subject. The loss of support might create a bodily response and an alarming aversive feeling associated with AI activation. At the same time, based on our finding showing AI activation is negatively associated with the feeling of belonging, we propose that this process creates a new “me and them” by increasing self-awareness or individuality [56,57]. This process might reduce the “group-self” feeling associated with the DMN and activate survival systems geared towards an individual. Compared to the aTPJ, which might be more sensitive to the social significance of the stimuli, the insula is sensitive to the affective value of stimuli [58,59].

VS is a part of SN [60] and has been linked to reward prediction error, signaling differences between expected and observed value at decision outcomes [61,62,63,64]. Compared to nonsocial rewarding stimuli, the ventral striatum has been more engaged in rewarding social stimuli [65,66]. In multi-round trust games, increased activity in the ventral striatum has been observed to be related to higher reward prediction error signals [67], suggesting that this region carries information about errors in reward prediction that enable estimations of the co-operative behavior of other partners [68,69]. Thus, our observation of increased striatal activity in the OP might be related to the failure of obtaining support (reward) that was predicted.

Although the task had not changed, subjects needed increased visual cognitive functioning to make better guesses (subjective cognitive load), which might reflect as high activity in the occipital and lingual areas at the later part of the game [70]. The increased demand for visual functions might contribute to reduced DMN activity [71]. Indeed, it was shown that the reduction of DMN activity correlated with the task’s difficulty [72].

The high numbers of squares and confounding structures in the task was designed to reduce the subjects’ confidence in their answers and become more dependent on their friends for wins. Through this approach, the social support by biased points was the main predictor for BOLD changes. The loss of support was accepted as a threat signal, which induced activation in the AI and aTPJ. The other factors that might be accepted as threat signals were loss of money or losing competition with motivation loss. However, VAS showed no difference between pre- and post-game motivations. We gave a symbolic amount of money to the groups, and interestingly, many groups reported enthusiasm for attending an fMRI study rather than obtaining money. However, these handicaps are limitations of our study. It should be noted that the number of subjects involved in this study was low despite the robustness of the results that increased the power of the study. Our sample consisted of young subjects with normal or high self-esteem which prevents our results from inferring to the general population. Furthermore, it should be kept in mind that low-frequency drift might be present in long fMRI trials or scans, which can persist over extended periods. This extended exposure to low-frequency noise poses an even greater potential for confounding and reduced signal-to-noise ratio, further hindering the accuracy and interpretability of the data. To overcome this problem, we limit our trial duration to 24 s (corresponding to a frequency of 0.04 Hz). In this way, we can effectively attenuate the impact of low-frequency drift while still capturing the neural dynamics of interest. Additionally, we examine the power spectrum of significant peak voxels within this restricted frequency range to enhance our ability to isolate and interpret meaningful neural signals while minimizing the influence of low-frequency noise.

## 5. Conclusions

In conclusion, the support from social groups provides an essential predictable environment for the multi-domain functioning of the DMN. On the other hand, loss of support as a social treat signal creates an unpredictable environment that activates the aTPJ and AI, which might play essential roles during switching from DMN to attention networks for survival against immediate environmental challenges.

## Figures and Tables

**Figure 1 brainsci-13-01509-f001:**
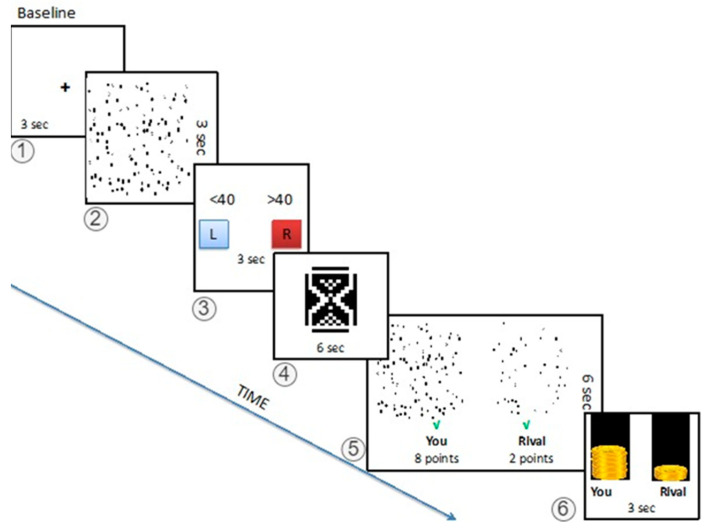
Task Outline. The task begins with a fixed cross for 3 s. Subjects are asked to guess the number of squares in 3 s in a picture with different numbers of various geometrical shapes. After providing their answer, the subjects see an hourglass for 6 s, indicating 10 points being distributed by the referees (anticipating stage). At stage 5, the subjects see both their own and the rivals’ screenshots, and the distribution of the points by the jury. A green checkmark indicates a correct answer, and a red cross indicates a wrong one (the example shows a biased response for the subject even though both competitors had correct answers). The last image shows the distribution of earnings.

**Figure 2 brainsci-13-01509-f002:**
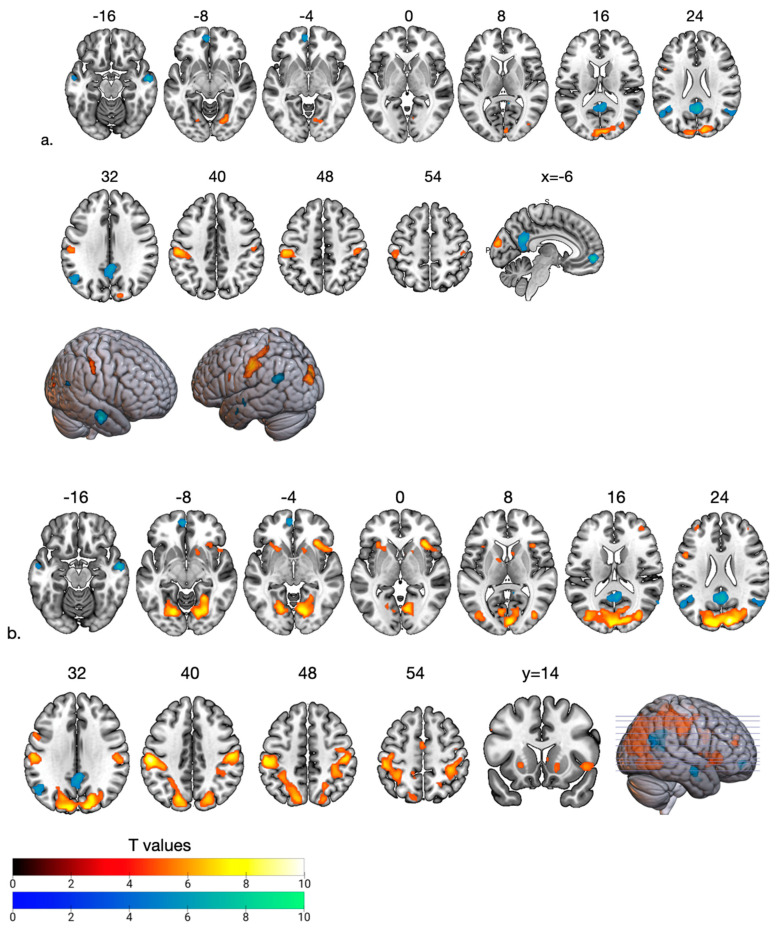
Hot and cool colors show increased and decreased BOLD signals during gradual social support loss at the fair (**a**) and ostracism phases (**b**) compared to HSP. The reduced activity in the DMN regions (mPFC, PCC, angular gyrus-posterior temporoparietal junction) and middle temporal cortex observed in the fair phase persisted in the ostracism phase. Gradual activity increase was observed in the supramarginal gyrus (anterior temporoparietal junction), dorsal parietal cortex, dlPFC (BA 10), and visual cortical areas with the loss of social support during fair and ostracism phases. Increased caudate/putamen nucleus (ventral striatum) and insula activity was observed during the ostracism phase. All images are neurological conventions. For illustrative purposes statistical maps are displayed with a threshold of *p* < 0.07 FEW and superimposed on a MNI template.

**Figure 3 brainsci-13-01509-f003:**
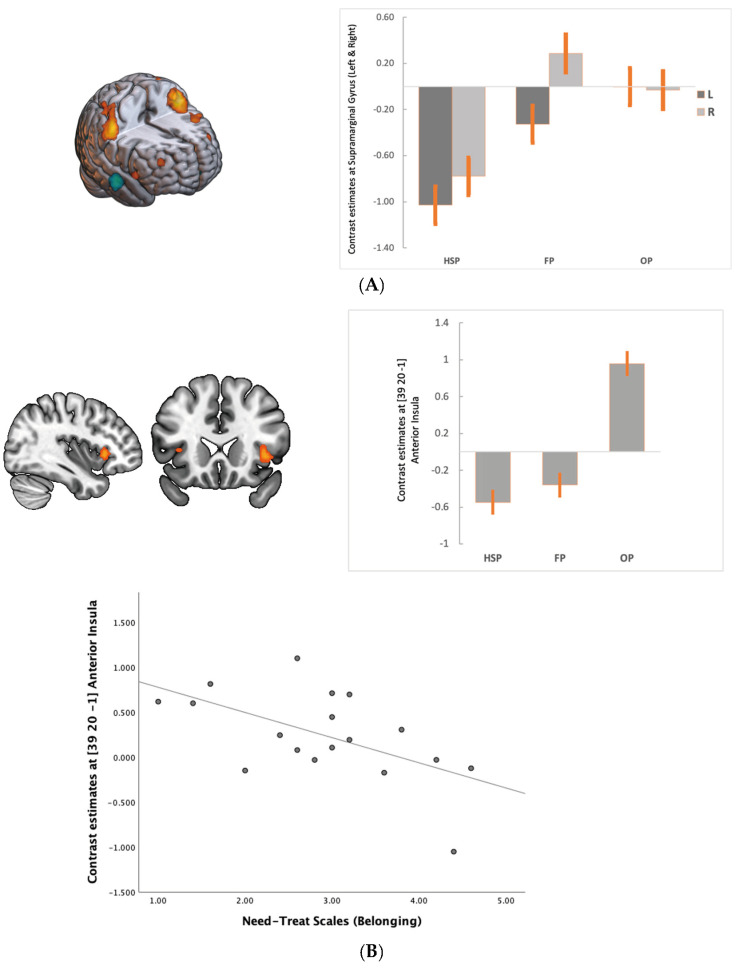
(**A**). Compared to HSP, we observed increased activity in the bilateral supramarginal gyrus in FP, which might be related to early environmental change in the game. (**B**). Anterior insula activity increased mainly with the social support loss in OP. Right anterior insula activity was negatively correlated with belonging to a group (r = −0.57 *p* < 0.05). For illustrative purposes statistical maps are displayed with a threshold of *p* < 0.07 FEW and superimposed on a MNI template. All images are neurological conventions.

**Table 1 brainsci-13-01509-t001:** The BOLD Signal Change During Fair Phase Compared to High Support Phase.

Area	Cluster Size	*p* (FWE)	T	x	y	z
Fair Phase > High Social Support						
R Supramarginal Gyrus (BA 40)	37	0.002	5.9	48	−22	47
L Supramarginal Gyrus (BA 40)	219	<0.001	7.8	−54	−25	44
R Visual Areas (BA 18)	240	<0.001	7.1	9	−91	20
R Visual Areas (BA 18)	34	<0.001	6.2	21	−73	−7
L dlPFC (BA 44)	8	0.016	5.4	−51	8	26
High Social Support > Fair Phase						
PCC (BA 23)	322	<0.001	8.8	3	−58	23
L mPFC (BA 10)	56	<0.001	8.3	−6	53	−4
L Angular Gyrus (BA 39)	84	<0.001	6.9	−54	64	29
R Angular Gyrus (BA 39)	8	0.031	5.2	51	−58	23
R middle temporal cortex (BA 21)	85	<0.001	8.5	57	−7	−16
L middle temporal cortex (BA 21)	6	0.043	5.3	−63	−14	−4

The regions that were not laterality labeled had clusters extend bilaterally R, Right; L, Left; BA, Brodmann Area; mPFC, medial prefrontal cortex; PCC, posterior cingulate cortex; dlPFC, dorsolateral prefrontal cortex.

**Table 2 brainsci-13-01509-t002:** The BOLD Signal Change During Ostracism Phase Compared to High Support Phase.

Area	Cluster Size	*p* (FWE)	T	x	y	z
Ostracism Phase > High Social Support						
L Supramarginal Gyrus (BA 40)	2384	<0.001	9.6	−57	−25	41
R Supramarginal Gyrus (BA 40)	358	<0.001	7.8	51	−19	41
R Insula (BA 13)	98	<0.001	7.5	34	23	−1
L Insula (BA 13)	41	<0.001	6.3	−33	20	2
R Visual Areas (BA 18)	34	<0.001	6.2	21	−73	−7
L Dorsolateral PFC (BA 44)	39	<0.001	6.3	−51	8	26
R dlPFC (BA10)	16	<0.001	6.3	42	47	17
L dlPFC (BA10)	8	<0.01	5.5	−33	47	26
R dPC (BA 7)	21	0.002	5.9	24	−58	50
L Premotor Areas (BA 6)	17	0.004	5.8	−3	−13	50
R Caudate	17	0.003	5.8	15	17	−7
R Caudate	8	0.006	5.6	9	11	5
L Putamen	7	0.005	5.8	−12	2	5
High Social Support > Ostracism Phase						
PCC (BA 23)	226	<0.001	7.11	3	−55	26
L mPFC (BA 10)	28	<0.001	6.8	−6	56	−7
L Angular Gyrus (BA 39)	92	<0.001	6.7	−43	58	29
R Angular Gyrus (BA 39)	24	0.013	5.2	54	−61	23
R middle temporal cortex (BA 21)	40	<0.001	7.4	57	−7	−16
L middle temporal cortex (BA 21)	30	0.043	6.3	−57	−7	−19

The regions that were not laterality labeled had clusters extend bilaterally R, Right; L, Left; BA, Brodmann Area; mPFC, medial prefrontal cortex; dPC, dorsal parietal cortex; PCC, posterior cingulate cortex; dlPFC, dorsolateral prefrontal cortex.

**Table 3 brainsci-13-01509-t003:** The BOLD Signal Change During Ostracism Phase Compared to Fair Phase.

Area	Cluster Size	*p* (FWE)	T	x	y	z
Fair Phase > Ostracism Phase						
*-----*						
Ostracism Phase > Fair Phase						
R Anterior Insula (BA 13)	39	<0.001	6.4	39	20	−1
L Anterior Insula (BA 13)	5	0.004	5.7	−36	−17	5
L Visual Area (BA 19)	89	<0.001	7.1	−13	−85	38
L Visual Area (BA 18)	16	0.003	5.8	−15	−73	−14
R Visual Area (BA 19)	18	0.004	5.7	18	−73	32

R, Right; L, Left; BA, Brodmann Area.

## Data Availability

We intend to share data following completion of all relevant study analyses and anticipate uploading raw data to NeuroVault (https://neurovault.org/, accessed on 8 October 2023) in late 2023. In the interim, it can reach upon request by contacting the corresponding author.

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
