# Peer review of "Gradual Loss of Social Group Support during Competition Activates Anterior TPJ and Insula but Deactivates Default Mode Network"

_brainsci, 2023, doi:10.3390/brainsci13111509_

Round 1

Reviewer 1 Report

The paper explored the neural correlates of losing social support. This is an overall interesting topic. And the author showed some interesting results. However, I have the following concerns for the authors to consider to improve the manuscript.

Introduction

1.     The authors in general did a good job in reviewing the related papers, but maybe better to restrict them a little to make the logic flow smoother.

Methods

1.     Line 253, condition is usually used for describing the experimental design, maybe stage is a better term to describe the task procedure. (Minor)

2.     Line 282, the task was designed to capture the crucial properties of losing social support, so the supporting level of the group was decreasing throughout the task. But how would the participants explain the decrease? As nothing happened, why would the group support decrease? Did the authors ask some debriefing questions? (Major)

3.     Line 291-292, what did the author mean by saying the behavioral variables were analyzed using number, percentage, mean, and standard deviation? Did they compare the standard deviation between phases? (Minor)

4.     Line 326, full-factorial analysis for within design is generally not recommended, the authors should use the flexible factorial design in SPM to replicate the results. (Major)

5.     For the method used in general, the authors compared fMRI scans far from each other (e.g. the HSP vs. OP), it is usually hard to dissociate the experimental effect from other confounds such as practice vs. fatigue, and machine drifting. Could the authors explain what they did to exclude potential alternative explanations? (Major)

6.     Line, 372, did the authors use a small volume correction? (Minor)

7.     Line, 400, Could the authors elaborate a little bit about how they did the connectivity analysis? Did they compute connectivity for each phase/stage?

Results

1.     Line 427, the author should use the term consistently, as here anticipation phase is a screen on a trial while the ‘phase’ has already been used in describing different task stages/blocks. (Minor)

The quality of the English can be improved, some of the sentences are hard to follow. 

Reviewer 2 Report

The manuscript “Social Group Support During Competition Activates Default Mode Network” aimed to examine neural activity associated with different types of social support (i.e., secure, unpredictable, insecure). Overall, I think this is an important topic for research, however, I think the authors could greatly strengthen the manuscript in several ways. My comments below outline my concerns.

1.     In the introduction the authors go into a detailed account of the fMRI task, I would suggest moving this information to the methods section and keeping the introduction primarily about the theoretical setup of the paper.

2.     Also, in the introduction the authors state in line 69 “In recent years, with the help of network approaches, we start…” I suggest revising this language as it seems to set up the idea that the authors are going to conduct a functional connectivity analysis, when they are really referring to multiple regions being associated with a functional process. Maybe saying something like “In recent years, with the help of meta-analyses and meta-analytic tools such as NeuroSynth, researchers have started to better understand how networks of brain regions map onto cognitive processes of interest.”

3.     Finally, in the introduction the authors set up the idea that social support involves regions associated with the default mode network, however, then go into processes related to self-processing, social processing or mentalizing, and conflict detection. It seems from a theoretical standpoint the paper would be strengthened if these specific cognitive processes were discussed in relationship to social support and dropping the reference to the default mode network.

4.     In the fMRI task, why did the authors present the social support conditions in order rather than in random order? Justification for this approach should be added to the task details.

5.     In the results section there is a lot going on and it is hard to follow. It seems to be a series of whole brain analyses. I’m wondering where the ROI and functional connectivity analyses are in the main text? Maybe I am just confused in what I am seeing. I also am seeing different p-values in the whole brain tables; the authors should select a single threshold and report all the findings for the one threshold.

6.     I realize some of this information noted above is in the supplemental materials, however, it currently reads more as here are a lot of findings across a bunch of different contrasts and I’m not understanding the importance. For example, why is it important to understand mechanisms associated with anticipation? I understand there are neural differences in activity, but how does this inform research on social support. This should all be set up in the introduction.  

7.     Back to my point at the beginning, I think this manuscript would be strengthened by connecting theorized processes with the contrasts of interest. For example, feedback stage seems important for social support, but reporting everything gets confusing. I would also suggest organizing the results in the main text to first report the ROI analyses, then PPI analyses, and finally exploratory whole brain. For the ROI analyses my suggestion would be to use NeuroSynth ROIs for the search terms “self-referential”, “mentalizing”, and “conflict”. As an example, maybe consider looking at paired t-tests to compare activity in these networks during (supportive feedback > baseline) versus (unsupportive feedback > baseline).

8.     Overall, I think refining the analysis a bit may yield results that can move the paper beyond the DMN would be helpful given the DMN is so vast and involves multiple processes. In addition, reorganizing the introduction and discussion in the same way would help with clarity and connecting the findings with previous literature and making more of a theoretical contribution.      

I think the language could be strengthened a bit to help with clarity, however, nothing major noted.  

Round 2
